# Effect of fetal malposition, primiparous, and premature rupture of membrane on Neonatal Near miss mediated by grade three meconium-stained amniotic fluids and duration of the active first stage of labor: Mediation analysis

**Mengstu Melkamu Asaye**[1]*, **Kassahun Alemu Gelaye**[2], **Yohannes Hailu Matebe**[3], **Helena Lindgren**[4], **Kerstin Erlandsson**[4,5]

1 Department of Women and Family Health, School of Midwifery, College of Medicine and Health Sciences, University of Gondar, Gondar, Ethiopia, 2 Department of Epidemiology and Biostatistics, Institute of Public Health, College of Medicine and Health Sciences, University of Gondar, Gondar, Ethiopia, 3 Department of Pediatrics and Child Health, School of Medicine, College of Medicine and Health Sciences, University of Gondar, Gondar, Ethiopia, 4 Department of Women's and Children's Health, Karolinska Institute, Solna, Sweden, 5 Institution for Health and Welfare, Dalarna University, Solna, Sweden

* mengstum@gmail.com

## Abstract

### Background

In many low-income countries, including Ethiopia, neonatal mortality remains a major concern. For every newborn that dies, many more neonates survived (near-miss neonates) the first 28 days after birth from life-threatening conditions. The generation of evidence on neonatal near-miss determinants could be a critical step in reducing neonatal mortality rates. However, studies causal pathway determinants are limited in Ethiopia. This study aimed to investigate the Neonatal Near-miss determinants in public health hospitals in Amhara Regional State, northwest Ethiopia.

### Method

A cross-sectional study was conducted on 1277 mother-newborn pairs at six hospitals between July 2021 and January 2022. A validated interviewer-administered questionnaire and a review of medical records were used to collect data. Data were entered into Epi-Info version 7.1.2 and exported to STATA version 16 in California, America for analysis. The paths from exposure variables to Neonatal Near-Miss via mediators were examined using multiple logistic regression analysis. The adjusted odds ratio (AOR) and ß-coefficients were calculated and reported with a 95% confidence interval and a p-value of 0.05.

### Results

The proportion of neonatal near-misses was 28.6% (365/1277) (95% CI: 26–31%). Women who could not read and write (AOR = 1.67,95%CI:1.14–2.47), being primiparous (AOR =

**Data Availability Statement:** All relevant data are within the paper and its Supporting Information files.

**Funding:** The study was funded by University of Gondar with reference number of (6-9/03/2013). However, the founder had no role in the design of the study, data collection, analysis, in writing of the manuscript and decision to publish.

**Competing interests:** The authors declared that there are no competing interests.

2.48,95% CI:1.63–3.79), pregnancy-induced hypertension (AOR = 2.10,95% CI:1.49–2.95), being referred from other health facilities (AOR = 2.28,95% CI:1.88–3.29), premature rupture of membrane (AOR = 1.47,95% CI:1.09–1.98), and fetal malposition (AOR = 1.89,95% CI:1.14–3.16) were associated with Neonatal Near-miss. Grade III meconium stained amniotic fluid partially mediated the relationship between primiparous (ß = 0.517), fetal malposition (ß = 0.526), pregnant women referred from other health facilities (ß = 0.948) and Neonatal Near-Miss at P-value < 0.01. Duration of the active first stage of labour partially mediated the relationship between primiparous (ß = -0.345), fetal malposition (ß = -0.656), premature rupture of membranes (ß = -0.550) and Neonatal Near-Miss at P- value <0.01.It had also a significant indirect effect (ß = 0.581, P<0.001) on NNM with variables (primiparous, fetal malposition, and premature rupture of membranes).

## Conclusions

The relationship between fetal malposition, primiparous, referred from other health facilities, premature rupture of membrane, and Neonatal Near miss were partially mediated by grade III meconium stained amniotic fluid and duration of the active first stage of labour. Early diagnosis of these potential danger signs and appropriate intervention could be of supreme importance in reducing NNM.

## Background

Neonatal mortality remains a major problem in many low-resource countries [1], and for every newborn who dies, there are many more newborns who survive from a life-threatening condition. The concept of Neonatal Near-Miss is not well understood [2]. It arose from the Maternal Near Miss concept [3]. Neonatal Near Miss has a varied magnitude around the world. The proportion was 22% in northwestern Brazil [4], 17.2% in Australia [5], 86% in central Gujarat, India [6], and 79% in Nepal [7] in facility based studies. In three Africa countries like Benin, Burkina Faso, and Morocco, it was 27.1%, 19.1%, and 30.4%, respectively [3], and 36.7% in central Uganda [8].

Ethiopia set a goal of reducing newborn mortality to 11 per 1000 live births in 2019/2020 [9]. However, in 2019, it surpassed 30 per 1000 live births [10]. Ethiopia's Amhara region has the highest rate (47 per 1000 live births) [11]. Neonatal Near Miss also varies in Ethiopia which was 26.7% in Jima zone [12], 32.2% in Debre Tabor [13], 33.4% in Hawasa city hospitals [21], and 45.1% in the southwest of the country [14].

Pregnant women who were referred from other health facilities [6, 15, 16], pregnancy-induced hypertension [12, 13, 17, 18], primiparous [13, 19, 20], premature rupture of membrane [13, 19, 21], low educational status [22, 23], fetal malposition [24, 25], birth interval less than 24 months, and severe maternal complications in pregnancy increased the risk of NNM [8, 14] and all of these factors contribute to the poor perinatal outcomes as well [26–28].

Grade three meconium-stained amniotic fluids is dense and have a "pea soup" consistency when there is a small amount of amniotic fluid but a large amount of meconium [27]. The entry of meconium into the amniotic fluid inside the uterus is thought to cause fetal distress. It could be a toxin if the fetus aspirates the particles inside the uterus while gasping for air or taking its first breath after birth. Because of its link to increased perinatal mortality and morbidity, its detection during labour causes concern between delivery room midwives and obstetricians. It may also be troubling for the women and the fetus [26, 29].

The duration of cervical dilatation is used to assess labour progress. Slow labour progression is becoming more common in obstetrical practices, results in a longer active first stage of labour and poor birth outcomes [25]. It has been attributed to primiparous women [20, 25, 30], fetal malposition [24, 31], and early membranes rupture [32].

The near-miss concept could be useful in newborn settings for identifying quality-of-care issues and improving health-care systems. An audit and feedback system will improve the quality of care and the performance of health care providers by evaluating medical care events that may lead to life-threatening situations [33, 34]. We may shift from failure to success and learn how to save lives in life-threatening situations if we focus on near-miss neonates and provide them with high-quality care [3, 17, 35]. The NNM approach would help in the detection and correction of newborn care mistakes. It can serve as a reminder to health care providers that newborns who have survived life-threatening situations should be monitored and followed up [2]. Similarly, it can also be used to inform policymakers on how to effectively use limited resources to improve quality of care [36].

In Ethiopia, no previous study utilized a validated neonatal near-miss assessment scale (NNMAS) to determine the magnitude of neonatal near-miss (NNM). However, we recently developed a new contextualized, validated, and reliable NNM assessment scale (NNMAS), which was tested in Ethiopia [37, 38]. Obstetric variables have a causal effect on Neonatal Near miss. Identifying and quantifying the causal pathway through mediation analysis could help in evidence generation and specific intervention planning. However, the mechanism by which these obstetric variables influence Neonatal Near misses has not been studied. Thus, the study aimed to investigate neonatal near-miss determinants in Amhara Regional state public hospitals, where the NNM scale was first validated [37, 38].

## Methods and materials

### Study design, setting, and period

Between July 2021 and February 2022, a cross-sectional study was conducted on live-birth neonates in six public health hospitals in the Amhara Regional state of northwest Ethiopia. Among the hospitals chosen were the University of Gondar comprehensive specialized hospital, Debretabor general hospital, and Debark, Gaynt primary hospital, Debre Markos referral hospital, and Felegehiwot comprehensive specialized hospital. Every maternity ward has triage, follow-up, second stages, and postnatal units. The newborn ward is also divided into sections. Each ward had senior doctors, residents, midwives, and nurses. The average number of monthly birth rate ranged from 135 to 340 [38]. Annually, of 13,640 deliveries at maternity wards, 2340 neonates were admitted to neonatal units. Data on live born newborns and medical records were collected. The medical records of five newborns were incomplete.

### Study population

The study included all singleton and live birth newborns. The study excluded twins, stillbirths, home births, and readmitted neonates.

### Study size and sampling

The sample size was determined using a single population proportion formula for reproducibility with the following statistical assumptions: proportion of neonatal near-misses was 41.5% [38], a level of significance of 5%, a Z/2 of 1.96, and a margin of error of 4%. The ultimate sample size was 1282 after considering design effects of 2 and a 10% non-response rate.

Six public health hospitals were selected randomly using a lottery method. The calculated sample size was proportionally allocated to each hospital based on their previous six-month reports. The number of sampling interval was four. Using a lottery method, the first case in each hospital was selected at random. Finally, using a systematic random sampling technique, live-born newborns were selected.

## Variables and data collection procedures

The dependent variable was Neonatal Near-Miss. While socio-demographic factors (age, place of residence, religion, marital status, maternal and husband educational status, maternal and husband occupation, number of families, and monthly income) and obstetric-related factors (antenatal care, number of antenatal care, iron supplementation, abortion, parity, ante-partum hemorrhage, antenatal care hemoglobin determination, pregnancy intention, history of still-birth, neonatal death, pregnancy-induced hypertension, Premature rupture of membranes, being referred from other health facilities, fetal malposition, grade III meconium stained amniotic fluid, duration of active first stage of labour and mode of delivery) were independent variables.

Data were collected on mother-newborn pairs through interviewer-administered questionnaire and medical record reviews. A validated [37] and reliable Neonatal Near-Miss Assessment Scale (NNMAS) was used [38]. Nine research assistants and three supervisors were involved in the data collection. Both research assistants and supervisors received two days of training. The data collection process was monitored daily and the acquired data were reviewed for accuracy and consistency. One computer was used for data entry.

## Data processing and analysis

We used a mediation analysis to examine the mechanism by which one independent variable influences the dependent variable via mediator variables to investigate a relationship. Neonatal Near-miss was the outcome (Y), and (grade III meconium-stained amniotic fluid and duration of the active first stage of labour) were identified as mediators (M).We used three regression analysis procedures to create this mediation analysis model.The first step was to find the effect (referred to as C) of X on Y(X➡Y; M-unadjusted);the second step was to obtain the effect (referred to as a) of X on M(X➡M) and the third step was to simultaneously determine the effect (referred to as b) of the M;M on Y(M–Y);X-adjusted) and the effect (referred to as C') of the X;X on Y (X➡Y; M-adjusted) [39] (**Fig 1**).

The mediation effect is established when the following three conditions are met: 1) there is a significant total effect (c). 2) A significant indirect effect (ab), and 3) the direct effect (c') is smaller than the total effect (c). (i.e. $|c'|<|c|$. If the direct effect is still significant, it is interpreted as "partially mediated" and if not "completely mediated" on the P-value of 0.05 for each path [40].The total effect(C) of the exposure on the outcome that is mediated by an intermediary variable can be estimated as [ab+ c'),or c-c'] [41].

Data were manually checked, coded, and entered into Epi-Info 7.1.2. For data validation, cleaning, and analysis; the data were exported to STATA version16. Categorical variables were reported using frequency and proportions. Bivariate logistic regression was used to identify the exposure variables associated with NNM and mediators. Exposure factors with a P-value of less than 0.2 were included in multivariable logistic regressions to account for confounding variables. To evaluate model fitness, the Hosmer-Lemshow goodness of fit (P = 0.47) was applied. The absence of multicollinearity among independent variables was determined when the variance inflation factor was less than 10. There was no multicollinearity among independent variables.

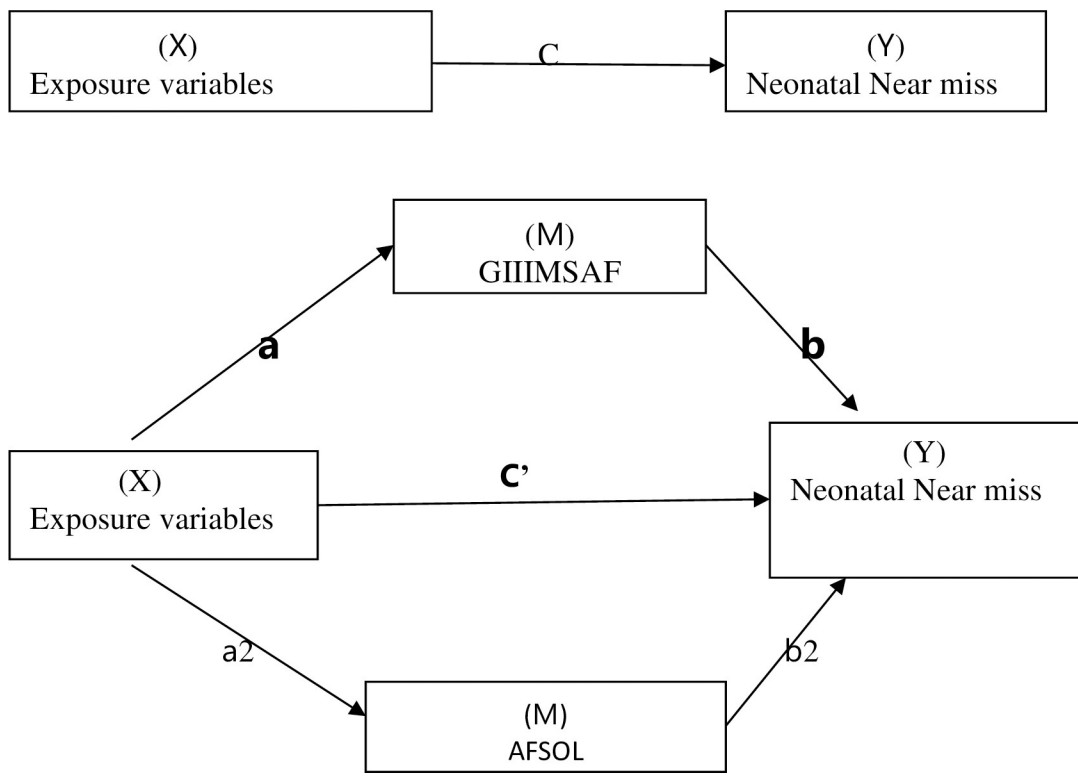

**Fig 1. This is the Neonatal Near Misses mediation model had three methods for evaluating the total effect (c), indirect effect (ab), and direct effect (c').** Premature rupture of membranes, being referred from other health facilities, fetal malposition and being primiparous were the exposure variables.

The relationship between independent factors, two mediators (duration of active first stage of labour and grade III meconium-stained amniotic fluid) and NNM was studied using multiple logistic regression and path analysis. The results were reported as odds ratios (ORs) and ß-coefficients with confidence intervals of 95% and P- values<0.05.

## Ethical approval

The study received ethical approval from the University of Gondar's Institutional Review Board (IRB) (Ref.No. V/P/RCSC/05/2543/2021. Support letters were sent to each hospital's administrative staff, and permission was obtained. Informed written consent was obtained from newborn mothers after a thorough explanation of the study's purpose, potential risks, benefits, and confidentiality. No personal identifiable information was recorded. The questionnaires, as well as all data collected from newborn mothers and reviews, were kept in a secure location and used solely for research purposes.

## Operational definitions

Neonatal Near Miss (NNM) is defined as neonates with severe complications who nearly died but survived immediately after birth in selected hospitals labour wards or on the first day of admission to a neonatal intensive care unit after being born in such hospitals, and who met at least one of the validated Neonatal Near-Miss Assessment Scale criteria [38].

Pregnancy-Induced Hypertension (PIH) was defined as a new onset of hypertension that arises after 20 weeks of pregnancy and elevated blood pressure (systolic≥140 or diastolic ≥90mmHg).

Premature rupture of membranes (PROM) is rupture of the membranes or leakage of amniotic fluid before labour begins but after 28 completed weeks of gestational age.

Birth Interval is the time between two most recent consecutive births or the duration of months between the birth of the index child and the subsequent birth.

Fetal malposition is defined as the occiput of the fetus in vertex presentation rotating in the occipital transverse or posterior position of maternal pelvis [24].

Grade III Meconium stained Amniotic Fluid (GIIIMSAF) is the passage of meconium by the fetus in the uterus with a high amount of meconium and little amniotic fluid; the staining is very thick during labour [27].

Prolonged active first stage of labor (AFSOL) is defined as the time interval between the latent stage of labor and full cervical dilation that is greater than or equal to 8 hours in both parity types.

## Results

### Socio-demographic characteristics

With a response rate of 99.6%, 1277 mother- newborn pairs were successfully interviewed and their medical records evaluated. The mothers of the babies were on average 28 years old (±5.5). Most of the survey participants, 919 (72%) lived in urban residency. The majority of the neonates' mothers, 1247(97.7%) were married, and 1260 (98.6%) were Orthodox Christians. Almost a quarter, 291(22.8%) of the newborn mothers could not read or write. Mothers of babies who had a college education or above constituted 371(29%) of cases, 302 (24%) were government employed, while 449 (35.2%) had a monthly income that fell within the poor group (Table 1).

### Obstetrics related characteristics

In this study, 1186 (93%) of the women received antenatal care, with 258 (21.8%) having fewer than four antenatal contacts. A total of 1107 (86.7%) of the mothers received iron and foliate supplements during pregnancy. The results also revealed that 568 (44.5%) of the mothers were primiparous, while 1106 (86.6%) had a planned pregnancy. Short birth intervals were observed in 160 (22.6%) of the mothers, while 37 (2.9%) had stillbirths and 13 (1%) had neonatal deaths. The haemoglobin levels during pregnancy were unknown in 112 (8.8%) of the mothers. Among the study participants, 284 women representing 22.2% had premature rupture of the membranes (PROM), 14.3% experienced pregnancy induced hypertension (PIH) while 683 (53.5%) were referred to the hospitals from other health facilities. The results also showed that 499 (39.1%) of the study participants completed their active first stage of labour within 8 hours or less, 78 (6.1%) encountered fetal malposition during labour and birth, 91% had grade three meconium-stained amniotic fluids, while 568 (44.5%) gave birth via caesarean section (Table 2).

### Neonatal near-miss and its determinants

The proportion of neonatal near-misses was 28.6% (365/1277) (95% CI: 26–31%). Women who could not read and write, being primiparous, pregnancy-induced hypertension, pregnant women referred from other health facilities, premature rupture of membranes, and fetal malposition were all associated with neonatal near misses.

Premature ruptures of membranes, being primiparous and fetal malposition were significantly associated with the duration of the active first stage of labour and NNM. Similarly, being primiparous, pregnant women referred from other health facilities, and fetal malposition were all significantly associated with grade three meconium-stained amniotic fluid and NNM.

**Table 1. Study participants' socio-demographic characteristics in public hospitals in Amhara Regional State, Northwest Ethiopia in 2021 (n = 1277).**

| Variables | Frequency | Percentage |
|---|---|---|
| Age | | |
| Less than or equal to20 | 101 | 7.9 |
| 21–35 | 1,045 | 81.8 |
| Greater than 35 | 131 | 10.3 |
| Marital status | | |
| Married | 1,247 | 97.7 |
| Others* | 30 | 2.4 |
| Religion | | |
| Orthodox | 1,231 | 96.4 |
| Other** | 46 | 3.6 |
| Ethnicity | | |
| Amhara | 1260 | 98.6 |
| Other*** | 17 | 1.4 |
| Maternal educational status | | |
| Could not read and write | 291 | 22.8 |
| Read and write | 197 | 15.4 |
| Primary school | 197 | 15.4 |
| Secondary school | 221 | 17.3 |
| College and above | 371 | 29.1 |
| Husband educational status | | |
| Unable to read and write | 238 | 18.6 |
| Read and write | 207 | 16.2 |
| Primary school | 160 | 12.5 |
| Secondary school | 196 | 15.4 |
| College and above | 476 | 37.3 |
| Maternal occupation | | |
| Government employed | 302 | 23.6 |
| House-wife | 480 | 37.6 |
| Farmer | 226 | 17.7 |
| Merchant | 148 | 11.6 |
| Non-government employed | 41 | 3.2 |
| Other*** | 80 | 6.3 |
| Husband occupation | | |
| Government employed | 437 | 34.2 |
| Non-government employed | 104 | 8.1 |
| Farmer | 366 | 28.7 |
| Merchant | 248 | 19.4 |
| Other**** | 122 | 9.6 |
| Average family monthly income | | |
| Poor | 449 | 35.2 |
| Medium | 408 | 32.0 |
| Rich | 420 | 32.9 |

NB:

* divorced, widowed, and single

**Muslim and protestant

**** Qimant and Agew

***daily labor, student and driver

**Table 2. Study participants' obstetric characteristics in public hospitals in Amhara Regional State, Northwest Ethiopia in 2021 (n = 1277).**

| Variables | Frequency | Percentage |
|---|---|---|
| ANC service utilization | | |
| Yes | 1186 | 93 |
| No | 91 | 7 |
| Number of ANC contact (n = 1186) | | |
| Less than 4 | 258 | 21.8 |
| 4 and above | 928 | 78.2 |
| Iron-foliate supplementation | | |
| Yes | 1107 | 86.7 |
| No | 170 | 13.3 |
| Birth interval (n = 709) | | |
| <24 months | 160 | 22.6 |
| ≥24 months | 549 | 77.4 |
| Parity level | | |
| 1 | 568 | 44.5 |
| 2–3 | 494 | 38.7 |
| 4 and above | 215 | 16.8 |
| Was it a planned pregnancy? | | |
| Yes | 1106 | 86.6 |
| No | 171 | 13.4 |
| History of stillbirth | | |
| Yes | 37 | 2.9 |
| No | 1,240 | 97.1 |
| History of neonatal death | | |
| Yes | 13 | 1 |
| No | 1264 | 99 |
| Was her hemoglobin level known at antenatal care? | | |
| Yes | 1165 | 91.2 |
| No | 112 | 8.8 |
| Was there a premature rupture of the membrane? | | |
| Yes | 284 | 22.2 |
| No | 993 | 77.8 |
| Was there pregnancy-induced hypertension | | |
| Yes | 182 | 14.3 |
| No | 1095 | 85.7 |
| Mode of admission to this hospital | | |
| Self | 594 | 46.5 |
| Referral | 683 | 53.5 |
| Was there malposition in labor? | | |
| Yes | 78 | 6.1 |
| No | 1199 | 93.9 |
| Duration of the active first stage of labor? | | |
| Less than 8 hours | 778 | 60.9 |
| Greater than or equal to 8 hours | 499 | 39.1 |
| Was there grade III meconium-stained amniotic fluid? | | |
| Yes | 116 | 9.1 |
| No | 1161 | 90.9 |

(*Continued*)

**Table 2.** (Continued)

| Variables | Frequency | Percentage |
|---|---|---|
| What was the mode of birth? | | |
| Spontaneous vaginal | 660 | 51.7 |
| Instrumental | 49 | 3.8 |
| Cesarean section | 568 | 44.5 |

Neonatal Near Miss was 1.67 times more common in neonates born to mothers who could not read and write (AOR = 1.67, 95% CI: 1.14–2.47) than in babies born to women with a college diploma or higher. The risks of NNM were 2.48 times higher (AOR = 2.48, 95% CI: 1.63–3.79) in newborns born to primiparous mothers compared to newborns born to women with four or more parity levels. The chances of NNM were 2 times higher among neonates born to women who had pregnancy-induced hypertension as compared to those who did not have PIH (AOR = 2.10, 95% CI: 1.49–2.95).

The odds of NNM were 2.28 times higher (AOR = 2.28, 95% CI: 1.88–3.28) among neonates born to mothers who were referred from other health facilities to the study hospitals compared to women who were not referred. The risks of NNM were 1.47 times higher (AOR = 1.47, 95% CI: 1.09–1.98) in neonates born to women who had a premature rupture of the membrane compared to babies born to women who did not have a premature rupture. The chances of NNM were 1.89 times higher (AOR = 1.89, 95% CI: 1.14–3.16) in neonates born to women who had fetal malposition during labour and birth compared to those delivered to women who did not have fetal malposition during labour and birth.

When compared to their peers, the risks of NNM among neonates born by mothers with grade three meconium-stained amniotic fluid (GIIIMSAF) during delivery were 1.97 times higher (AOR = 1.97, 95% CI:1.21–3.19). When comparing neonates born by women who had an active first stage of labour lasting more than 8 hours to those born by women who had an active first stage of labour for less than 8 hours, the odds of NNM were 1.81 times higher (AOR = 1.81, 95% CI: 1.41–2.31) (Table 3).

The study showed that the duration of AFSOL and GIIIMSAF were the obstetric factors that mediated NNM. Women who were primiparous (AOR = 3.85, 95% CI: 1.81–8.11), women referred from other health facilities (AOR = 2.81, 95% CI: 1.76–4.48), and women who had fetal malposition during labour (AOR = 2.08, 95% CI: 1.07–4.02) were all associated with grade III meconium stained amniotic fluid.

When compared to women with 4 or more parity levels, the odds of AFSOL less than 8 hours among the primiparous mothers were 49% lower (AOR = 0.51, 95% CI: 0.36–0.71). Those who had premature rupture of the membrane were 1.51 times more likely to experience active first stage of labour (AFSOL) with less than 8 hours (AOR = 1.51, 95% CI: 1.14–2.01) than mothers who did not have a premature rupture of the membrane (AOR = 1.51, 95% CI: 1.14–2.01).

Mothers of new-borns who did not have fetal malposition during labour and birth were 44% less likely to have AFSOL in less than 8 hours than those who had fetal malposition (AOR = 0.56, 95% CI: 0.34–0.91) (Table 3)

## The mediation effect of grade III meconium stained amniotic fluid and active first stage of labour on Neonatal Near-miss

A three-step logistic regression analysis was employed to determine the total effect (c), indirect effect (ab), and direct effect (c'),.Premature rupture of membranes, primiparous, women

**Table 3. The mediation effect of grade III meconium stained amniotic fluid and active first stage of labour on the relationship between obstetric variables and Neonatal Near Miss in public hospitals in Amhara Regional state, Northwest Ethiopia 2021 (n = 1277).**

| Variables | Model I(estimating c) | | Model II(Estimating a) | | | | Model III (estimating b and c') | | | | Mediation type |
|---|---|---|---|---|---|---|---|---|---|---|---|
| | Neonatal Near-Miss | | GIIIMSAF | | AFSOL | | Neonatal Near-Miss | | | | |
| | AOR95%CI | ß | AOR95%CI | ß | AOR95%CI | ß | AOR95%CI | ß | AOR95%CI | ß | |
| Parity level | | | | | | | | | | | |
| 1 | 2.48(1.63–3.79) | 1.930* | 3.85(1.83–8.11) | 1.241* | 0.51(0.36–0.71) | 0.660* | 1.68(1.15–2.44) | 0.517* | 0.71(0.59–0.84) | -0.345* | P" |
| 2–3 | 1.46(0.95–2.24) | 0.194 | 1.30(0.59–2.88) | 0.906 | 0.75(0.52–1.06) | 0.176 | 1.14(0.76–1.69) | 0.129 | 0.80(0.67–1.46) | -0.005 | _ |
| 4 and above | 1 | ref. | 1 | ref. | 1 | ref. | 1 | ref. | 1 | ref. | |
| Mode of admission | | | | | | | | | | | |
| Self | 1 | ref. | 1 | ref. | 1 | ref. | 1 | ref. | | | |
| Referral | 2.28(1.88–3.28) | 2.238* | 2.81(1.76–4.48) | 1.125* | 0.86(0.67–1.09) | -0.201 | 2.58(1.97–3.38) | 0.949* | | | P |
| Fetal malposition | | | | | | | | | | | |
| No | 1 | ref. | 1 | ref. | 1 | ref. | 1 | ref. | 1 | ref. | |
| Yes | 1.89(1.14–3.16) | 0.930** | 2.08(1.07–4.02) | 0.857** | 0.56(0.34–0.91) | -0.530* | 1.69(1.02–2.81) | 0.526* | 0.52(0.32–0.85) | -0.656* | P" |
| PROM | | | | | | | | | | | |
| No | 1 | ref. | 1 | ref. | 1 | ref. | | | 1 | ref. | |
| Yes | 1.47(1.09–1.98) | 0.523* | 1.04(0.66–1.64) | 0.219 | 1.51(1.14–2.01) | 0.752* | | | 1.74(1.40–2.32) | 0.550* | P |
| Maternal education | | | | | | | | | | | |
| Unable to read and write | 1.67(1.14–2.47) | 0.538** | 1.11(0.61–2.03) | 0.038 | 1.01(0.70–1.43) | 0.150 | | | | | N |
| Read and Write | 0.85(0.62–1.25) | -0.051 | 0.68(0.35–1.32) | -0.135 | 0.73(0.51–1.05) | -0.334 | | | | | _ |
| Primary school | 0.95(0.62–1.43) | 0.084 | 1.26(0.71–2.25) | 0.432 | 1.19(0.83–1.74) | 0.107 | | | | | _ |
| Secondary school | 1.29(0.87–1.91) | 0.319 | 0.73(0.39–1.36) | -0.063 | 0.95(0.67–1.34) | -0.109 | | | | | _ |
| College and above | 1 | ref. | 1 | ref. | 1 | ref. | | | | | |
| Pregnancy-induced htn | | | | | | | | | | | |
| No | 1 | ref | 1 | ref. | 1 | ref. | | | | | |
| Yes | 2.10(1.49–2.95) | 0.802* | 0.58(0.31–1.11) | -0.396 | 0.86(0.62–1.18) | -0.130 | | | | | N |
| **GIIIMSAF** | | | | | | | | | | | |
| No | | | | | | | 1 | ref. | | | |
| Yes | | | | | | | 3.15(2.08–4.74) | 1.146 | | | |
| **AFSOL** | | | | | | | | | | | |
| ≥8hours | | | | | | | | | 1.78(1.38–2.30) | 0.581* | |
| <8hours | | | | | | | | | 1 | ref. | |

**GIIIMSAF**: grade three meconium-stained amniotic fluid; **AFSOL**: the active first stage of labor; **AOR**: adjusted odds ratio; **CI**: confidence interval; **PROM**: premature rupture of membrane: standardized coefficients

*P<0.001 and

**p<0.01.

Capital letters indicate the type of mediation based on the degree of effect (- = no relation; **N** = no mediation; **P** = partial mediation on one mediator and **P"** = partial mediation on two mediators)

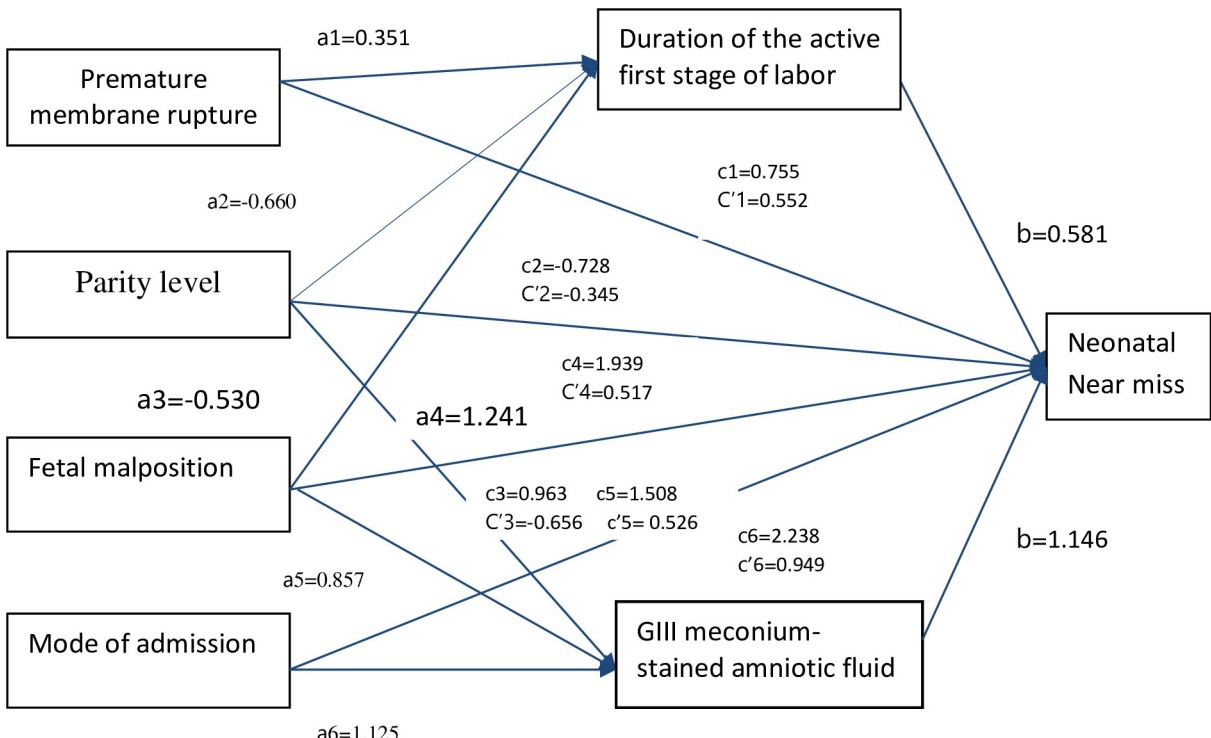

**Fig 2. This is the mediation analysis of path coefficients that was used to estimate the total effect (c), indirect effect (ab), and direct effect (c') on Neonatal Near Miss in public hospitals in Amhara Regional state, Northwest Ethiopia 2021 (n = 1277).**

referred from other facilities, and fetal malposition during labour all had a significant effect on NNM. The paths from the above-listed covariates to NNM were mediated by the duration of the active first stage of labour and grade three meconium-stained amniotic fluid, as evidenced by the direct effects being smaller than the total effects $|c'| < |c|$ (**Fig 2**).

We found a significant relationship between fetal malposition during labour and the active first stage of labour lasting less than 8 hours (ß = -0.530, P<0.001) and NNM (ß = 0.961, P<0.01).Similarly, it showed a positive and significant relationship with grade three meconium-stained amniotic fluid (ß = 0.857, P<0.01) and NNM (ß = 1.504, P<0.01).The study predicted a positive and significant relationship between premature rupture of membranes and the active first stage of labour lasting less than 8 hours (ß = 0.351, P<0.05) and NNM (ß = 0.752, P<0.001).The study also revealed that being primiparous had a significant relationship with AFSOL lasting less than 8 hours (ß = -0.066, P<0.001) and NNM (ß = -0.726, P<0.001). Primiparity also had a positive and significant direct relationship with GIIIMSAF (ß = 1.241, P<0.001) and NNM (ß = 1.938, P<0.001). There was a positive and significant direct relationship between pregnant women referred from other health facilities and the likelihood of grade three meconium-stained amniotic fluid level (ß = 1.125, P<0.001) and NNM (ß = 2.238, P<0.001).

Grade three meconium-stained amniotic fluid partially mediated the relationship between primiparous (ß = 0.517, P<0.001), fetal malposition (ß = 0.526, P<0.001), pregnant women referred from other health facilities (ß = 0.948, P<0.01) and NNM. The duration of the active first stage of labour also partially mediated the relationship between primiparous (ß = -0.345, P<0.001), fetal malposition (ß = -0.656, P<0.001), premature rupture of membranes (ß = -0.550, P<0.01) and NNM.

The mediator grade three meconium-stained amniotic fluid (ß = 1.146, P<0.001) had a significant indirect effect on NNM in neonates born to women with fetal malposition during labour, pregnant women referred from other health facilities, and primiparous women. The duration of the active first stage of labour (ß = 0.581, P<0.001) had a significant indirect effect on NNM in neonates born to mothers who had fetal malposition, premature rupture of membranes, and primiparous (Table 3).

## Discussion

The study found a high proportion of Neonatal Near Misses. The relationship between fetal malposition, primiparous, being referred from another health facility, premature rupture of membrane, and Neonatal Near Miss was partially mediated by grade III meconium stained amniotic fluid and active first stage of labour duration.

The study revealed that the proportion of Neonatal Near-miss was 28.6%. In comparison to studies conducted in other health facilities in Northeastern Brazil [4], Australia [5], and Burkina Faso [3], the proportion was higher. This could be due to a disparity in the quality of care provided in the health facilities in these countries. In addition, prospective follow-up was used in the Brazil and Australia studies, and the outcome was determined using a different measurement scale, whereas in our study, we used a contextualized [37], valid, and reliable NNMAS [38].This finding, on the other hand, was consistent with other studies from Brazil [43], and the Jima Zone, southwest Ethiopia [12].

The NNM proportion of 28.6% in our study was, however, lower when compared to other studies conducted in health facilities in Brazil [42], central Gujarat India [6], Nepal [7], Central Uganda [8], Hawasa Ethiopia [43], South Ethiopia [14], and Debre Tabor hospital, northwest Ethiopia [13]. Participants in these other studies in Brazil, central Gujarat India, and Uganda, were neonates born to women hospitalized with severe obstetric problems.This could be one of the contributing factors to the disparity. It could be related to the study locations chosen. Studies in Nepal, Uganda, and Debre Tabor were conducted on single high-caseload referral hospitals. This could explain the high number of neonatal near miss cases. It could also be due to different scales used to measure outcome. Thus, studies in central Gujarat-India, used only pragmatic criteria to determine outcome [6], whereas studies in Brazil used mechanical ventilation and congenital malformation characteristics with a pragmatic scale to determine outcome [42].

In contrast, in the studies conducted in Nepal, Hawasa-Ethiopia, and Debretabor Hospital in Northwest Ethiopia, results were measured using a combination of pragmatic and management criteria [7, 13, 43]. The current study was conducted at all levels of hospitals. We used a contextualized [37], and validated NNMAS for outcome ascertainment [38]. This would increase the magnitude variations.

In our study, newborns born to women who could not read and write were 1.67 times more likely to develop NNM than those born to mothers who had a college education or higher. It was supported by a study conducted in Italy and Germany, which stated that education is the most powerful health determinant, and that low educational levels worsen poor birth outcomes [22, 23].

The findings in our study show that the chances of having NNM were 2.48 times higher among newborns born to primiparous women than newborns born to mothers with four or more parity levels. This is consistent with the findings in a study conducted at Debre Tabor Hospital in Northwest Ethiopia [13], and tends to suggest that psychological concerns, stress, and intense labour pain may have a significant impact on primiparous women's energy and power during labour and birth. This may result in prolonged labour which has negative

consequences for the newborn [20]. So, while being multiparous is risky to the mother, the infant being born to a primiparous mother is even at a greater risk. As a result, during labour and birth, midwives must pay special attention to these vulnerable groups.

In this study, newborns born to women who had pregnancy-induced hypertension were twice as likely as those born to mothers who did not have pregnancy-induced hypertension to have NNM. This finding is similar to the findings reported in studies conducted in Brazil [17], Ethiopia's Jima Zone hospitals [12], and Debretabor hospital in Northwest Ethiopia [13]. This could be attributed to intrauterine growth retardation (IUGR), placental insufficiency, and increased interventional delivery that may be performed regardless of the gestational age. As a result, low birth weight and preterm newborns are more likely [18]. Thus, midwives must provide additional assistance and attention to those mothers.

In the current study, babies born to women who were referred from other health facilities were 2.28 times more likely to experience NNM than those who were not referred. This finding is consistent with studies conducted in Ethiopia's Gurage zone hospitals [15],and central Gujarat-India [44]. This could be due to a lack of equipment and qualified personnel at local health facilities that provide delivery services. As a result, obstetric emergencies are delayed. Thus, the babies would be vulnerable to adverse birth outcomes [16]. It is, therefore, recommended that midwives should prioritize providing timely treatment to those vulnerable new-borns.

The study revealed that babies born to women who had a premature rupture of the membrane were 1.47 times more likely to develop NNM than those mothers who did not have PROM. The finding is in agreement with those reported in Ethiopia's Gamo-Gofa Zone [19], and Debre Tabor Hospital [13]. As a result, when the amniotic membrane ruptures early, the risk of infection increases. It also increases the chance of having a premature or early-born child [21].

Evidence suggests that neonates born in fetal malposition are more likely to have low APGAR scores, delivery trauma [24], and poor birth outcomes [25]. In line with this, our study showed that neonates born to women who had fetal malposition during labour and birth were 1.89 times more likely to develop NNM than those without fetal malposition. This finding should serve as a guide to midwives to spot the danger early and call for immediate action to save the lives of the new-borns.

The current study found that neonates born to women who had grade III meconium-stained amniotic fluid (GIIIMSAF) during labour had 1.97-fold increase in the risk of NNM. This finding is consistent with a study in India which found that highly thick meconium-stained amniotic fluid is a sign of fetal distress [29]. As a result, early detection and intervention are required to reduce the risk of high-level meconium-stained amniotic fluid. It would help to a decrease in newborn morbidity [26, 28].

A prolonged active first stage of labour has been linked to an increased risk of adverse newborn outcomes [25]. This is consistent with the findings in our study which showed that neonates born to women who had AFSOL lasting more than 8 hours had a 1.81-fold increased risk of NNM than those born to women who had AFSOL lasting less than 8 hours. This may be due to the increased likelihood of fetal distress and a low APGAR score if labour lasts longer. The most important strategy for early detection of prolonged AFSOL is the consistent use and interpretation of the partograph.

Newborns born to primiparous mothers had a lower chance of completing the active first stage of labour in less than 8 hours, increasing the risk of GIIIMSAF and NNM. Being primiparous was associated with an active first stage of labor lasting less than 8 hours (ß = -0.660, P<0.01), grade three meconium-stained amniotic fluid (ß = 1.241, P<0.01), and NNM (β = 1.930, P<0.01). It was partially mediated by GIIIMSAF (ß = 0.517) and AFSOL (ß = -0.345) with NNM at P<0.01. Neonatal Near-miss [13], labor elongation due to stress [20], and grade

three meconium-stained amniotic fluids have all been reported in primiparous mothers. A previous study supported up this claim [29].

It was also found that neonates born to women who had fetal malposition during labour and birth had a lower chance of going through an active first stage of labour in less than 8 hours, increasing the chances of GIIIMSAF and NNM. It had a significant relationship with the duration of the active first stage of labour being less than 8 hours (ß = -0.531), grade three meconium-stained amniotic fluid (ß = 0.857), and NNM (ß = 0.930) at P<0.05. It was partially mediated by GIIIMSAF (ß = 0.526) and AFSOL (ß = -0.656) with NNM at P<0.01. According to the Portland study [25], fetal malposition causes delayed labor, meconium-stained amniotic fluid [24], and poor delivery outcomes. It has an indirect effect on the NNM via grade three meconium-stained amniotic fluid (ß = 1.241),which is supported by previous studies [27, 29] and (ß = 0.581) active first stage of labour [25], as mediators at P<0.01.

Neonates born to mothers who had a premature rupture of the membrane would complete the active first stage of labour in less than 8 hours, but at a higher risk of NNM. Premature rupture of membrane was partially mediated by AFSOL (ß = -0.550) with NNM at p<0.001.Early detection and prompt antibiotic treatment are critical to avoid sepsis. This finding is supported by other studies conducted in Ethiopia [13, 19]. Premature rupture of membrane has an indirect effect on NNM via AFSOL (β = 0.581, P<0.001) of the mediator. While early membrane rupture can reduce labour time, it also increases the risk of infection and poor birth outcomes [32].

The likelihood of having grade three meconium-stained amniotic fluid (ß = 1.125, P<0.001) and the number of NNM newborns (ß = 2.238, P<0.001) were both positive and significant. It had been partially mediated by GIIIMSAF (ß = 0.949, P<0.001) with NNM. Prior studies support this finding [6, 15]. Women referred from other health facilities have an indirect effect on NNM via mediator grade three meconium-stained amniotic fluid (ß = 1.146, P<0.001). It could be related to obstetrics emergency delays and the vulnerability of neonates to poor outcomes [16].

The strength of this study lies in the fact that it is a multi-centre study with a large sample size. In addition, a psychometrically validated scale was used to assess the outcome. The study also determined the direct and indirect effects of variables on NNM via mediators. The data were collected over a specific time from women-newborn pairs as well as their medical records at various hospitals. As a limitation, women who gave birth to live neonates at home or elsewhere but were referred to one of the selected hospitals were excluded and only live births were studied in the early neonatal period with no follow-up. We strongly recommend that a study be conducted to cover the entire neonatal period.

The study could aid in the early detection and timely management of obstetric danger signs that could endanger a baby's life, as well serve as a reminder to midwives involved in the follow-up and labour monitoring of pregnant women who exhibit risk factors. Midwives should look for meconium-stained amniotic fluid, a prolonged active first stage of labour, and fetal malposition, which could indicate a moribund new-born. It can also be used educate midwives, doctors and other health professionals on how to manage limited resources to improve quality of care and reduce NNM and mortality.

## Conclusion

The proportion of Neonatal Near-Miss was high. Women who could not read or write, primiparous, referred from a health facility, fetal malposition, premature rupture of membranes, and had pregnancy-induced hypertensions were all statistically significant risk factors for NNM. The relationship between fetal malposition, primiparous, referred from other health

facilities, premature rupture of membrane, and NNM were partially mediated by grade III meconium stained amniotic fluid and the duration of active first stage of labour. Early diagnosis and intervention of these potential danger signs could be of supreme importance in reducing NNM.

## Supporting information

**S1 Table. This is the relationship between maternal related characteristics and Neonatal Near miss.**
(DOCX)

**S2 Table. This is the relationship between obstetrics characterizes and duration of the active first stage of labor.**
(DOCX)

**S3 Table. This is the relationship between obstetrics characteristics and grade III meconium-stained amniotic fluid.**
(DOCX)

**S1 File. This is the SPSS dataset of the study.**
(SAV)

## Acknowledgments

The authors are very grateful to the University of Gondar for the approval of the ethical clearance. Our gratitude also goes to the Amhara Region Health Bureau for its supportive. Finally, we would like to extend our appreciation to the study participants, research assistants, and supervisors.

## Author Contributions

**Conceptualization:** Mengstu Melkamu Asaye.

**Data curation:** Mengstu Melkamu Asaye, Kassahun Alemu Gelaye, Yohannes Hailu Matebe, Helena Lindgren, Kerstin Erlandsson.

**Formal analysis:** Mengstu Melkamu Asaye, Kassahun Alemu Gelaye, Yohannes Hailu Matebe, Helena Lindgren, Kerstin Erlandsson.

**Funding acquisition:** Mengstu Melkamu Asaye.

**Investigation:** Mengstu Melkamu Asaye, Kassahun Alemu Gelaye, Yohannes Hailu Matebe, Helena Lindgren, Kerstin Erlandsson.

**Methodology:** Mengstu Melkamu Asaye, Kassahun Alemu Gelaye, Yohannes Hailu Matebe, Helena Lindgren, Kerstin Erlandsson.

**Project administration:** Mengstu Melkamu Asaye, Kassahun Alemu Gelaye, Yohannes Hailu Matebe, Helena Lindgren, Kerstin Erlandsson.

**Resources:** Mengstu Melkamu Asaye.

**Software:** Mengstu Melkamu Asaye, Kassahun Alemu Gelaye, Yohannes Hailu Matebe, Helena Lindgren, Kerstin Erlandsson.

**Supervision:** Mengstu Melkamu Asaye, Yohannes Hailu Matebe, Helena Lindgren, Kerstin Erlandsson.

**Validation:** Mengstu Melkamu Asaye, Kassahun Alemu Gelaye, Yohannes Hailu Matebe, Helena Lindgren, Kerstin Erlandsson.

**Visualization:** Mengstu Melkamu Asaye, Kassahun Alemu Gelaye, Yohannes Hailu Matebe, Helena Lindgren, Kerstin Erlandsson.

**Writing – original draft:** Mengstu Melkamu Asaye.

**Writing – review & editing:** Mengstu Melkamu Asaye, Kassahun Alemu Gelaye, Yohannes Hailu Matebe, Helena Lindgren, Kerstin Erlandsson.

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
