## [Decision Letter · Decision Letter 0]

22 Nov 2022

PONE-D-22-20623Effect of fetal malposition, primiparous, and premature rupture of membrane on Neonatal Near miss mediated by grade three meconium-stained amniotic fluids and duration of first stage of labour : Mediation analysisPLOS ONE

Dear Dr. Asaye,

Thank you for submitting your manuscript to PLOS ONE. After careful consideration, we feel that it has merit but does not fully meet PLOS ONE’s publication criteria as it currently stands. Therefore, we invite you to submit a revised version of the manuscript that addresses the points raised during the review process.

Your manuscript has been assessed by an expert reviewer, whose comments are appended below. The reviewer raised important concerns about some aspects of the methodology and presentation, which you should address carefully in your revised manuscript. Kindly address each point carefully in your response to the reviewers’ document and revise the manuscript accordingly.

Additionally, ensure you proofread and edit the manuscript for typographical and grammatical errors throughout the manuscript.

Line 36, write the correct meaning of the abbreviation AOR. Kindly note AOR means (adjusted odds ratio).

Line 38- ‘The proportion of neonatal near-misses was 28.6%’, In addition to the percentage add the count or number for instance 26% (X/Y). In the abstract, state the statistical software you used in analysing the data as well as the version, city, and country of the manufacturer.

Lines 41&42 -Correct punctuations

Line 98- the sentence is incomplete kindly revise

Line 122 - ‘The sample size was determined using a single population proportion formula.’ state the formula for reproducibility

Revise lines 127 to 130 by stating your sampling method first and then how it was carried out -for clarity. Line 128 ‘The number of skips was four’ do you mean the sampling interval?

Line 142 -use ‘research assistants’ rather than ‘data collectors’

In Line 190 kindly revise the sentence. Also, edit lines 203 and 249 for grammar and punctuation amongst others.

We look forward to receiving your revised manuscript.

Kind regards,

Felix Chikaike Clement Wekere

Academic Editor

PLOS ONE

Journal Requirements:

a) Did participants provide their written or verbal informed consent to participate in this study?

Reviewers' comments:

Reviewer's Responses to Questions

**Comments to the Author**

1. Is the manuscript technically sound, and do the data support the conclusions?

Reviewer #1: Yes

2. Has the statistical analysis been performed appropriately and rigorously? 

Reviewer #1: Yes

3. Have the authors made all data underlying the findings in their manuscript fully available?

Reviewer #1: Yes

4. Is the manuscript presented in an intelligible fashion and written in standard English?

Reviewer #1: No

5. Review Comments to the Author

Reviewer #1: ABSTRACT : The information in the abstract was well written. The background and methods section were well captured.

However under the results section, only the overall proportion of neonatal near- misses was captured included the AOR of the various determinants. It will be worth mentioning was percentages were attributed to the various obstetric determinants.

Background

‘The proportion was 22% in northwestern Brazil [4], 17.2% in Australia [5], 86% in central 4 65 Gujarat, India [6], and 79% in Nepal [7].In three Africa countries like Benin, Burkina Faso, and 66 Morocco, it was 27.1%, 19.1%, and 30.4%, respectively [3], and 36.7% in central Uganda [8]’………..Are the various proportions mentioned nationwide values or in selected facilities? Its unclear…..Kindly clarify.

Paragraph 4: ‘The entry of meconium into the amniotic fluid inside the uterus is thought to cause fetal distress by causing fetal hypoxia’…….fetal distress from fetal hypoxia causes fetus to pass meconium and not vice versa… meconium passage doesn’t cause fetal hypoxia… Kindly correct that

Operational definitions:

Kindly review the grammar …’ Neonatal Near Miss (NNM) is defined as when newborns with severe complications who 182 nearly died but survived at birth and who faced to at least one of the Neonatal Near Miss 183 Assessment scale (NNMAS) elements is referred to as a Neonatal Near Miss ‘

Discussion

Paragraph 3: ‘This finding was also lower when compared to studies conducted in Brazil [44], central 358 Gujarat- India [6],Nepal ……’ Which finding are you making reference to? Kindly clarify and ensure

Kindly review your grammar throughout the work as some sentences in your discussion begin with ‘ because’

6. PLOS authors have the option to publish the peer review history of their article (what does this mean?). If published, this will include your full peer review and any attached files.

Reviewer #1: No

---

## [Author Response · Author response to Decision Letter 0]

1 Dec 2022

Dear Editor. We sincerely appreciate your efforts and encouragements to produce a high-quality scientific paper for the scientific community. We have a high regard for your journal (PLOS ONE) because of the emphasis you placed on quality and scientifically sound articles. We attempted to respond to all of your comments in the main document and described in the rebuttal letter.

Dear Editor, I appreciate your detailed comments and prompt response. The manuscript has been reviewed for grammatical and punctuation errors in each line and page, and necessary changes have been made in the main document.

Dear reviewer, first and foremost, please accepts my heartfelt gratitude for your insightful feedback and comments. You brought up an extremely important point. We wanted to see the overall association through adjusted odds ratio rather than percentage of each attribute in the cross-sectional study. We attempted to respond to all of your comments in the main document and described in the rebuttal letter.

---

## [Decision Letter · Decision Letter 1]

30 Mar 2023

PONE-D-22-20623R1Effect of fetal malposition, primiparous, and premature rupture of membrane on Neonatal Near miss mediated by grade three meconium-stained amniotic fluids and duration of the active first stage of labor: Mediation analysisPLOS ONE

Dear Dr. Asaye,

Thank you for submitting your manuscript to PLOS ONE. After careful consideration, we feel that it has merit but does not fully meet PLOS ONE’s publication criteria as it currently stands. Therefore, we invite you to submit a revised version of the manuscript that addresses the points raised during the review process.

Thank you once again for resubmitting your manuscript for review. Your manuscript has been assessed by expert reviewers whose comments are appended below. The reviewers have raised important concerns about some aspects of the methodology and presentation, which you should address carefully in your revised manuscript. Kindly address each point carefully in your response to the reviewers’ document and revise the manuscript accordingly.Additionally, ensure you proofread and edit the manuscript for syntax and grammatical errors throughout the manuscript. Please, read through the review comments and suggestions made painstakingly and make the corrections to improve your manuscript.

We look forward to receiving your revised manuscript.

Kind regards,

Felix Chikaike Clement Wekere

Academic Editor

PLOS ONE

Journal Requirements:

Reviewers' comments:

Reviewer's Responses to Questions

**Comments to the Author**

1. If the authors have adequately addressed your comments raised in a previous round of review and you feel that this manuscript is now acceptable for publication, you may indicate that here to bypass the “Comments to the Author” section, enter your conflict of interest statement in the “Confidential to Editor” section, and submit your "Accept" recommendation.

Reviewer #2: All comments have been addressed

Reviewer #3: (No Response)

2. Is the manuscript technically sound, and do the data support the conclusions?

Reviewer #2: Yes

Reviewer #3: Yes

3. Has the statistical analysis been performed appropriately and rigorously? 

Reviewer #2: Yes

Reviewer #3: Yes

4. Have the authors made all data underlying the findings in their manuscript fully available?

Reviewer #2: Yes

Reviewer #3: Yes

5. Is the manuscript presented in an intelligible fashion and written in standard English?

Reviewer #2: Yes

Reviewer #3: Yes

6. Review Comments to the Author

Reviewer #2: On the whole, the study is a good one as it tried to highlight some of the major contributors to Neonatal Near Misses some of which contribute significantly to both maternal and perinatal mortality if not managed properly, especially in low-income resource areas. The typographical, grammatical and spelling errors have been highlighted for ease of reference and correction.

1. Abstract: Background:

a. Page 2: Lines 29: …… limited in Ethiopia

2. Abstract: Methods:

a. Page 2: line 34: ……and a review of medical records were used to….

b. Page 2: line 36: …. via mediators were examined

c. Page 2: line 37: ....and β-coefficients were calculated…

d. Page 3: lines 56-57: to read “Early diagnosis of these potential danger signs and appropriate intervention could be of supreme importance in reducing NNM”.

e. Page 4: line 69: delete “was” after Ethiopia.

f. Page 5: line 97: delete “on” after followed up.

g. Page 5: line 107: delete “at” after investigate.

3. Methods and Materials: Study design, setting and period:

a. Page 6: Line 116: to read: …the newborn ward “is” also divided…

b. Page 6: Line 117: midwife to read “Midwives”.

c. Page 6: Line 118-119: The sentence “Of these, 15,000 deliveries at maternity wards” does not make sense and should be reviewed! Are the Authors saying that there were 15,000 deliveries in all the maternity wards used for the study?

d. Page 6: Line 120: delete “born to women”! to read “The medical records of five newborns were incomplete”.

4. Variables and data collection procedures:

a. Page 7: Line 145: To read “three supervisors “were”

b. Page 7: Line 146-147: To read: “The data collection process was monitored daily and the acquired data were reviewed for accuracy and consistency. One computer was used for data entry.

c. Page 8: Line 165: ….and entered “into” Epi-Info 7.1.2.

d. Page 8: Line 172: inflation factor was less than 10.

5. Ethical Approval:

a. Page 9: Line 1179: “Support letter” to read “Support letters”

b. Page 9: Line 183: “as well as….

c. Page 9: Line 190: PIH was defined as a new onset of hypertension…

6. Results:

a. Page 10: Line 209: “couldn’t” to read: “could not”

b. Page 10: Line 210-212: to read: Mothers of babies who had a college education or above constituted 371 (29%) of cases, 302 (24%) were government employed, while 449 (35.2%) had a monthly income that fell within the poor group (Table 1).

c. Table 1: Page 12: Average family monthly income was classified as: Poor, Medium and Reach. It is not clear what the word “Reach” means or implies!

d. Page 226 under Table 1 NB: **Muslim and protestant and…(Please review)

7. Obstetrics related characteristics:

a. Page 12: Line 228-240: to read: In this study, 1186 (93%) of the women received antenatal care, with 258 (21.8%) having fewer than four antenatal contacts. A total of 1107 (86.7%) of the mothers received iron and foliate supplements during pregnancy. The results also revealed that 568 (44.5%) of the mothers were primiparous, while 1106 (86.6%) had a planned pregnancy. Short birth intervals were observed in 160 (22.6%) of the mothers, while 37 (2.9%) had stillbirths and 13 (1%) had neonatal deaths. The hemoglobin levels during pregnancy were unknown in 112 (8.8%) of the mothers. Among the study participants, 284 women representing 22.2% had premature rupture of the membranes (PROM), 14.3% experienced pregnancy induced hypertension (PIH) while 683 (53.5%) were referred to the hospitals from other health facilities. The results also showed that 499 (39.1%) of the study participants completed their active first stage of labour within 8 hours or less, 78 (6.1%) encountered fetal malposition during labour and birth, 91% had grade three meconium-stained amniotic fluids, while 568 (44.5%) gave birth via caesarean section (Table 2).

8. Neonatal near-miss and its determinants:

a. Page 14: line 256: “couldn’t” to read “could not”

b. Page 14: line 258: “associated to” to read “associated with”

c. Page 14: line 259: …….fetal malposition “were” significantly…

d. Page 14: line 260: “associated to the mediator duration of the AFSOL”… to read “associated with” the duration of the AFSOL….

e. Page 14: line 262: delete “mediator” before grade 3….

f. Page 14: line 268: to read …………as compared to those who did not have PIH.

g. Page 14: line 273: “a preterm rupture” to read “a premature rupture”

h. Page 15: line 284-285: review to read “the study showed that the duration of AFSOL and G111MSAF were the obstetric factors that mediated NNM”.

i. Page 15: line 291: delete “neonates’” and replace with “the” primiparous mothers….

j. Page 15: line 292: replace “early rupture” with “premature rupture”

k. Page 15/16: line 295-297: delete “when compared to” to read “Mothers of newborns who did not have fetal malposition during labour and birth were 44% less likely to have AFSOL in less than 8 hours than those who had fetal malposition (AOR = 0.56, 95% CI: 0.34-0.91). (Table 3)

l. Page 16: line 312: study also revealed that being primiparous….

m. Page 16: line 313: …. “Primiparous had also a positive”… to read “Primiparity also had a positive”…

9. Discussion:

a. Page 19: line 356: The study revealed that…..

b. Page 19 line 357: to read: “studies conducted in other health facilities in Northeastern Brazil…..

c. Page 19: line 358: to read: “This cause could be due to a disparity in the quality of care provided in the health facilities in these countries”. Delete to individuals on the verge of death but survived newborns

d. Page 19: line 359-361: to read “In addition, prospective follow-up was used in the Brazil and Australia studies, and the outcome was determined using a different measurement scale, whereas in our study, we used a contextualized [38}, valid, and reliable NNMAS [39].

e. Page 19: line 362: …….”was consistent with other studies from Brazil [46] and the Jima Zone, Southwest Ethiopia [12]”

f. Page 19: line: 364: to read: “The NNM rate of 28.6% in our study was, however, lower when compared to other studies conducted in health facilities in Brazil [44], central Gujarat India [6], Nepal [7] and Central Uganda [45}, Hawasa Ethiopia [46], south Ethiopia [14], and Debre Tabor hospital, Northwest Ethiopia [13]. Study participants Participants in these other studies in Brazil, central Gujarat India, and Uganda, were neonates born to women hospitalized with severe obstetric problems.

g. Page 20: line: 369: Add “were” after Debre Tabor.

h. Page 20: line: 375: to read: “In contrast, in the studies conducted in Nepal……..

i. Page 20: line: 380: In this finding our study, newborns born to women who couldn’t could not…..

j. Page 20: line 382: According to this study, The findings in our study show that the chances …….

k. Page 20: line 387-391: to read: This is consistent with the findings in a study conducted at Debre Tabor Hospital in Northwest Ethiopia [13], and tends to suggest that psychological concerns, stress, and intense labour pains may have a significant impact on primiparous women’s energy and power during labour and birth. This may result in prolonged labour which has negative consequences for the newborn [20]. So, while being multiparous is risky to the mother, the infant being born to a primiparous mother is even at a greater risk.

l. Page 21: line 395: to their counterparts read: ….This finding is similar to the findings reported in studies conducted in Brazil [17],……….

m. Page 21: line 396-397: to read: ……. This could be because attributed to intrauterine growth retardation (IUGR), placental insufficiency, and increased interventional delivery that may be performed regardless of the gestational age.

n. Page 21: line 402: to read: ……… than those who were not referred.

o. Page 20: line 406: to rea: Based on this It is, therefore, recommended that midwives should prioritize providing timely treatment to those vulnerable newborns.

p. Page 21: line 408: to read: The study suggested revealed that…….

q. Page 21: line 409: to read: ….. than their peers those mothers who did not have PROM….

r. Page 21: line 410: replace “accordance” with “agreement”

s. Age 22: line 416 -417: to read:…….than their counterparts those without fetal malposition. This is a warning to This finding should serve as a guide to midwives to spot it early and consult an obstetrician to take the danger early and call for immediate action to save the lives of the newborns.

t. Page 22: line 419: “1.97-fold increased” to read “1.97-fold increase” in the risk of NNM.

u. Line 420: delete “than their peers”

v. This finding was supported by” to read: This finding is consistent with a study in India which found that highly thick meconium-stained amniotic fluid is a sign of fetal distress [29]. Delete “which is concerning to the neonate’s better prognosis’

w. Page 22: line: 426: to read: …… This is consistent with the findings in our study which showed that neonates born to women who had AFSOL lasting more than 8 hours had a 1.81-fold increased risk of NNM than those born to women who had AFSOL lasting less than 8 hours. Delete than their counterparts

x. Page 22: line 428: to read: “This is” to read “This may be”…..

y. Page 22: line 429: to read: …….. “The most important strategy for intervening early detection of prolonged AFSOL is the consistent use and interpretation of the partograph.

z. Page 23: line 439: replace “endorses” with “supported”. Delete: “This is the responsibility of the midwives and obstetricians who work. It does not make any sense here!.

aa. Page 23 line 442: “lasting” to read “going through”

bb. Page 23: line 448: “indirectly” to read “indirect”

cc. Page 23: line 450: delete the whole sentence

dd. Page 23: line 454: Ethiopian studies supportsThis finding is supported by other studies conducted in Ethiopia [13, 19]

ee. Page 24: line 462: “this idea” to read “this finding”

ff. Page 24: line 466-467: to read: “The strength of this study lies in the fact that it is a multi-centre study with a large sample size. In addition, a psychometrically validated scale was used to assess the outcome. The study also determined the direct ………..

gg. Page 24: line 471-473: …….were “missed” excluded and only live births babies were studied in the early neonatal period with no follow-up. We strongly recommend that a study be conducted to cover the entire neonatal period.

hh. Page 25: line 478: replace: “nearly dead but living newborn” with “moribund newborn”.

ii. Page 25; line 479: to read: “educate midwives, doctors and other health professionals on how to manage limited resources in order to improve quality of care and reduce NNM and mortality.

10. Conclusion:

a. Page 25 line482: “couldn’t” to read “could not”

b. Page 25: line 488: ….. “could be of supreme importance in reducing NNM.”

Reviewer #3: Thank you for giving me the opportunity to review this interesting article that affirmed existing knowledge.

The authors rigorously statistical analysis of the data is highly commendable. However, there are repetitions of statements that makes the results and discussion pages very long. I suggest reduction of discussion pages by effective narration of results and correlation with other published work. Avoid repeating sentences already detailed in the results pages. A constructive discussion of results and not stating results in the discussion narrations.

The definition of prolonged active phase of labour used in this study may be justified further in relation to parity.

7. PLOS authors have the option to publish the peer review history of their article (what does this mean?). If published, this will include your full peer review and any attached files.

Reviewer #2: **Yes: **Dr Francis E. Alu

Consultant Obstetrician & Gynaecologist

Abuja Nigeria

Reviewer #3: No

---

## [Author Response · Author response to Decision Letter 1]

31 Mar 2023

Dear reviewers

Thank you for your detailed evaluation feedback, grammatical corrections, and insightful remarks. I really appreciated the time you took to improve our work. Your valuable and crucial feedback has taught me a lot. I hope the findings will benefit both the scientific community and clinical practice. Your comments have been incorporated into the attached point by point response letter as well as the documents attached. Thank you once more for your valuable time and contributions.

best regards

---

## [Editor Report · Decision Letter 2]

19 Apr 2023

Effect of fetal malposition, primiparous, and premature rupture of membrane on Neonatal Near miss mediated by grade three meconium-stained amniotic fluids and duration of the active first stage of labor: Mediation analysis

PONE-D-22-20623R2

Dear Dr. Asaye,

We’re pleased to inform you that your manuscript has been judged scientifically suitable for publication and will be formally accepted for publication once it meets all outstanding technical requirements.

Kind regards,

Felix Chikaike Clement Wekere

Academic Editor

PLOS ONE
---

## [Editor Report · Acceptance letter]

27 Apr 2023

PONE-D-22-20623R2 

Effect of fetal malposition, primiparous, and premature rupture of membrane on Neonatal Near miss mediated by grade three meconium-stained amniotic fluids and duration of the active first stage of labor: Mediation analysis 

Dear Dr. Asaye:

I'm pleased to inform you that your manuscript has been deemed suitable for publication in PLOS ONE. Congratulations! Your manuscript is now with our production department. 

Kind regards, 

on behalf of

Dr. Felix Chikaike Clement Wekere 

Academic Editor

PLOS ONE